# Loss of *STING* in *parkin* mutant flies suppresses muscle defects and mitochondria damage

Andrew T. Moehlman[1,2], Gil Kanfer[1], Richard J. Youle[1]*

**1** Biochemistry Section, Surgical Neurology Branch, National Institute of Neurological Disorders and Stroke, National Institutes of Health, Bethesda, Maryland, United States of America, **2** Postdoctoral Research Associate Training Program, National Institute of General Medical Sciences, National Institutes of Health, Bethesda, Maryland, United States of America

* youler@ninds.nih.gov

**Data Availability Statement:** All relevant data are within the manuscript and its Supporting information files. For RNA-seq experiments, a complete processed dataset, including normalized counts, DESeq2 results and accompanying

## Abstract

The early pathogenesis and underlying molecular causes of motor neuron degeneration in Parkinson's Disease (PD) remains unresolved. In the model organism *Drosophila melanogaster*, loss of the early-onset PD gene *parkin* (the ortholog of human PRKN) results in impaired climbing ability, damage to the indirect flight muscles, and mitochondrial fragmentation with swelling. These stressed mitochondria have been proposed to activate innate immune pathways through release of damage associated molecular patterns (DAMPs). Parkin-mediated mitophagy is hypothesized to suppress mitochondrial damage and subsequent activation of the cGAS/STING innate immunity pathway, but the relevance of this interaction in the fly remains unresolved. Using a combination of genetics, immunoassays, and RNA sequencing, we investigated a potential role for STING in the onset of *parkin*-null phenotypes. Our findings demonstrate that loss of *Drosophila* STING in flies rescues the thorax muscle defects and the climbing ability of *parkin*[-/-] mutants. Loss of STING also suppresses the disrupted mitochondrial morphology in *parkin*[-/-] flight muscles, suggesting unexpected feedback of STING on mitochondria integrity or activation of a compensatory mitochondrial pathway. In the animals lacking both *parkin* and *sting*, PINK1 is activated and cell death pathways are suppressed. These findings support a unique, non-canonical role for Drosophila STING in the cellular and organismal response to mitochondria stress.

## Author summary

Neurodegenerative diseases such as Parkinson's Disease have been associated with excessive inflammation. The anti-viral immune regulators STING (STimulator of INterferon Genes) and NF-kB have been indicated in mammalian studies to respond to unmitigated mitochondria damage linked to mutations in quality control pathways such as the Parkinson's Disease-associated Pink1/Parkin pathway. In the well-characterized model system *Drosophila melanogaster* (fruit fly), mutants lacking either of these genes show physical deformities and movement defects caused by mitochondria damage and progressive neurodegeneration, yielding a powerful and efficient model to study the effects of blocking

statistics can be found in S2 Data File. The original RNA-seq read files (in FASTQ format) are available at NCBI Gene Expression Omnibus (GEO, GSE232950). Additional raw data files and original R code used for data analysis are available on Figshare (https://figshare.com/projects/Drosophila_STING_and_Parkin_Manuscript/157578).

**Funding:** This work was supported by the Intramural Research Program of the National Institute of Neurological Disorders and Stroke, 1ZIANS003123-11 to R.J.Y. This work was supported in part by a NIGMS Postdoctoral Research Associate Training (PRAT) fellowship, FI2GM138078-01, to A.T.M. The funders had no role in study design, data collection and analysis, decision to publish, or preparation of the manuscript.

**Competing interests:** The authors have declared that no competing interests exist.

the conserved STING pathway. We find that loss of *STING* prevents multiple phenotypes of the *parkin* mutant fly, by multiple complementary approaches. Further, this suppression extends to decreasing the severity of the mitochondria dysfunction normally seen in *parkin* mutant animals. Finally, assays of gene expression in animals lacking both *STING* and *parkin* reveal a significant increase in anti-stress enzymes, which have been previously linked to suppressing degeneration in *parkin* fly models.

## Introduction

Mutations in PINK1 and Parkin lead to early onset Parkinson's disease (PD). PINK1 is a kinase imported to mitochondria and degraded, unless shunted to the outer mitochondrial membrane when mitochondrial membrane potential is impaired [1–3]. Once stabilized on the outer mitochondrial membrane (OMM) PINK1 phosphorylates ubiquitin and the Parkin ubiquitin-like domain to recruit the E3 ligase Parkin to the mitochondria, which amplifies OMM protein ubiquitination [4–7]. This ubiquitination promotes recruitment of autophagy receptors and autophagy of damaged mitochondria [8–10]. Although the molecular mechanisms of PINK1 and Parkin are well-studied, how their absence leads to Parkinsonian phenotypes is less clear [11]. Mutations in PINK1 and Parkin do not lead to substantial or PD-related phenotypes in otherwise healthy mice [12,13]. However, *Drosophila melanogaster* mutants lacking either *pink1* or *parkin* (*park*) have severe phenotypes [14–18]. Mutants in either *park* or *pink1* lose flight muscle, undergo degeneration of certain dopaminergic neurons, and display locomotion and flight impairment. Notably, mitochondria in the indirect flight muscles are swollen and the elongated morphology is disrupted [14,18]. Depletion of mitochondria fusion genes or expression of genes regulating fission can rescue *park*⁻/⁻ phenotypes, supporting a role for mitochondrial dynamics in the pathophysiology of these mutant phenotypes [19–22].

Unhealthy mitochondria activate innate immune pathways through release of damage associated molecular patterns (DAMPs) such as mitochondrial DNA (mtDNA) [23–25]. Human PD patients lacking PINK1 or Parkin exhibit increased inflammation and increased serum mtDNA [26], consistent with work showing that flies lacking Parkin express higher levels of genes implicated in oxidative stress and immune responses [27]. Parkin-dependent mitophagy has been proposed to limit mitochondrial DAMP release and subsequent activation of the cGAS/STING innate immune pathway, which was previously examined in *PRKN*⁻/⁻ mice models [28]. However, this study utilized stress paradigms, as unstressed mice do not exhibit PD-like phenotypes, unlike flies. Thus, we explored the potential role of STING in *parkin*-null flies. During our study, a report indicated that loss of STING did not rescue Drosophila *parkin* mutant defects [29], which contrasted with our contemporaneous preliminary data. Herein we compared different strains of *parkin* mutant alleles and conclude that loss of STING activity suppresses the thorax muscle involution and the climbing defects of *parkin*⁻/⁻ mutants. Surprisingly, loss of STING also improves the disrupted mitochondrial morphology in *parkin*⁻/⁻ flight muscles, suggesting unexpected feedback of *Drosophila* STING on mitochondrial homeostasis.

## Results

### STING is necessary for muscle degeneration and climbing defects in *parkin* flies

Thorax indention and bent wing phenotypes in *parkin* mutant flies are indicative of underlying indirect flight muscle (IFM) defects and attributed to mitochondria dysfunction inducing

muscle apoptosis [14,19,30]. We generated flies harboring null alleles for *parkin* (*park$^{25}$*) [14] and *sting* (*sting$^{ARG5}$*) [31]. Analysis of these double knockout (DKO) flies demonstrated that loss of *sting* rescued both the thorax and wing phenotypes of the *parkin* mutant flies (Fig 1A–1C). We obtained the independently derived *park$^1$* mutant and backcrossed this allele into the *sting$^{ARG5}$* mutant background [15]. These flies also demonstrated reduced penetrance of the *parkin* phenotypes (Fig 1A–1C). For both backgrounds, the status of the *sting* and *park*-null alleles were scored based on the presence or absence of the balancer chromosomes and fly genotypes were routinely confirmed using PCR (S1 Fig). Both *park$^{25}$* and *park$^1$* homozygous flies demonstrate climbing defects, due to muscle degeneration and, later, age-dependent loss of dopaminergic neurons [14,15,32]. Using the negative geotaxis assay (Fig 1D), flies homozygous for *sting$^{ARG5}$* were assayed for climbing ability in *parkin* wild-type, *park$^{25}$*, and *park$^1$* backgrounds. Loss of *sting* alone had no effect on climbing ability in young (5–7 days-old) flies. For both *parkin* alleles, loss of *sting* suppressed the climbing defects of young *parkin* null adults (Fig 1D and S1 Video).

We confirmed the veracity of the *sting$^{ARG5}$* knockout allele by crossing *sting$^{ARG5}$* flies with flies containing a *sting* deficiency chromosome in the *park$^{25}$* mutant background (S1E Fig). Resulting progeny harboring one copy of *sting$^{ARG5}$* allele and the sting deletion displayed suppressed thorax and wing phenotypes in the homozygous *park$^{25}$* mutation (S1F–S1H Fig). These results support a necessary role for STING in progression of muscle degeneration of *parkin$^{-/-}$* flies. We also observed that loss of *sting* in the *pink1$^5$* or *pink1$^{B9}$* hemizygous mutant background rescued the severity of the thorax phenotypes only partially and to a lesser extent than in *parkin*-null flies (Fig 1E–1G). Pink1 has been reported to have multiple Parkin-independent interactions [21,33–35], and loss of STING may not affect these pathways, resulting in only minor suppression of the *pink1* thoracic muscle phenotypes.

To investigate our results on *parkin* that diverge from Lee et al. [29], we acquired the *sting$^{ARG5}$; park$^{25}$* line used in that study. We verified these animals with RT-qPCR and scored thorax and wing phenotypes in the homozygous *sting$^{ARG5}$; park$^{25}$* flies (S2 Fig). This line of *sting$^{ARG5}$; park$^{25}$* flies displayed minimal thorax indention phenotype but retained the *park$^{25}$* bent wing phenotype (S2C Fig). One explanation for the divergent results compared to Lee et al. [29] could be differences in the genetic background. Therefore the gifted *sting$^{ARG5}$; park$^{25}$* stock underwent eight generations of outcrossing to the *w$^{1118}$* stock followed by single-male fly crosses to a double balancer stock as described in detail in the Materials and Methods section. Resulting fly lines that retained the *sting$^{ARG5}$* allele and the *park$^{25}$* allele, as tested with PCR, were self-crossed to test the resulting homozygous progeny. In the outcrossed stocks, loss of *sting* suppressed both the thorax indentation and bent wings of the *parkin* flies, compared to *sting$^{ARG5}$/+(heterozygous); park$^{25}$(homozygous)* siblings (S2B and S2C Fig). Therefore, it appears that a yet-unknown background difference could contribute to the severity of the *park$^{25}$* phenotypes in *sting$^{ARG5}$* mutant flies.

## STING influences the underlying mitochondria pathology in *parkin* flies

Defects in *parkin*-null flies include disrupted mitochondrial morphology in indirect flight muscles (IFM) [14–17]. This has been linked to dysfunctional mitochondrial dynamics attributed both to blocking of mitochondrial autophagy [36,37] and to disruption of mitochondria fusion and fission dynamics [20,38]. To assess the mitochondrial health in the *parkin$^{-/-}$* and *sting$^{-/-}$* flies, we examined the IFM-associated mitochondria in thoraces of young flies (3–5 days post-eclosion) using Alexa Fluor488-labeled streptavidin to visualize mitochondria (Fig 2) [39,40]. As previously reported, *park$^{25}$* and *park$^1$* mutants possess disrupted morphology, with interrupted mitochondrial networks and the appearance of large swollen

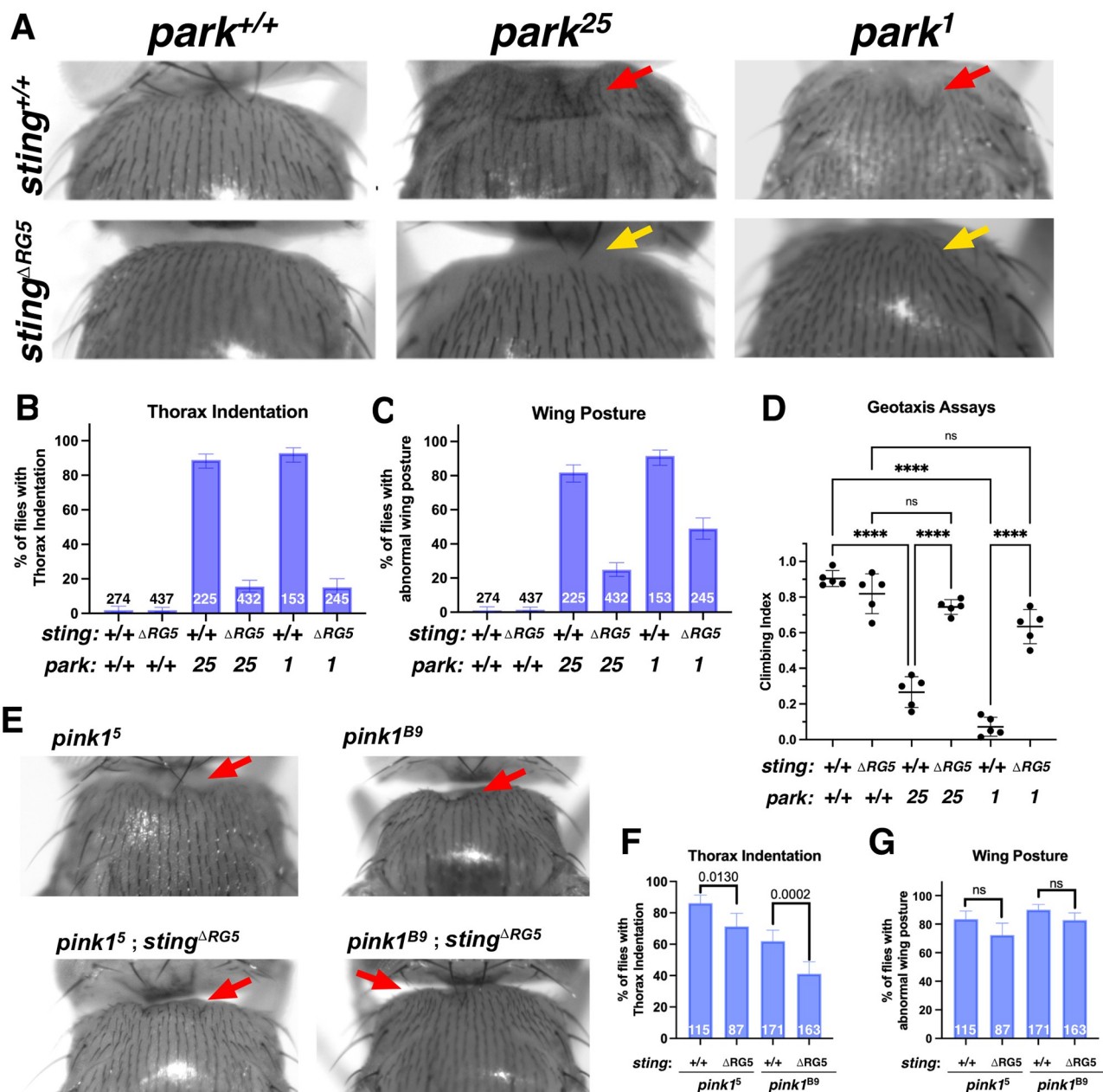

**Fig 1. STING mediates flight muscle degeneration in *parkin*<sup>-/-</sup> flies.** (**A**) Representative images of the thoracic muscle indentation. The *sting*<sup>ΔRG5</sup> allele crossed to either null *parkin* allele (yellow arrows) rescues the thoracic defects of *park*<sup>25</sup> and *park*<sup>1</sup> mutants (red arrow). All flies were generated in or crossed to a wild type *w*<sup>1118</sup> stock (**B & C**) Quantification of the thoracic indentations (B) or the downward bent wing posture (C) in the indicated genotypes. In all graphs, bars represent the percentage of flies displaying the indicated phenotype, numbers within or juxtaposed to the bars indicate the number of flies scored per genotype (n), and the error bars represent the 95% confidence interval for the population proportion. (**D**) Scatter plots of quantifications for negative geotaxis assays in the indicated genotypes. Each data point represents the mean of at least 3 technical replicate assays with a group of 15 to 20 flies. Horizontal bars indicate the mean of 5 independent biological replicas per genotype. Error bars display the standard deviation. Genotypes were tested for statistical significance with an 1-way ANOVA test with post-hoc multiple comparison testing with Bonferroni's correction. (**E**) Example images for *pink1*<sup>5</sup>, *pink1*<sup>B9</sup>, *pink1*<sup>5</sup>; *sting*<sup>ΔRG5</sup> and *pink1*<sup>B9</sup>; *sting*<sup>ΔRG5</sup> male flies. Note that the loss of *sting* slightly affects the pink1-null phenotypes, in contrast to the strong level of suppression seen in *parkin* mutant combinations. (**F & G**) Quantification of the thorax indention and wing posture defect phenotypes in *pink1*<sup>-/y</sup>, or *pink1*<sup>-/y</sup>; *sting*<sup>ΔRG5</sup> flies. In all graphs, bars represent the percentage of flies displaying the indicated phenotype, numbers indicate the number of flies scored per genotype, and the error bars represent the 95% confidence interval for the population proportion. Significance was determined using Fisher's Exact Test for differences between population proportions. Significant p-values are indicated on the graphs.

mitochondria aggregates (Fig 2C and 2E). These defects were substantially suppressed when the *sting*<sup>ΔRG5</sup> allele was crossed to either of the *park-null* alleles (Fig 2D and 2F), whereas loss of *sting* alone had no mitochondrial disruption compared to controls (Fig 2A and 2B). Blinded scoring of the IFM mitochondria integrity in randomized examples of ten thoraces

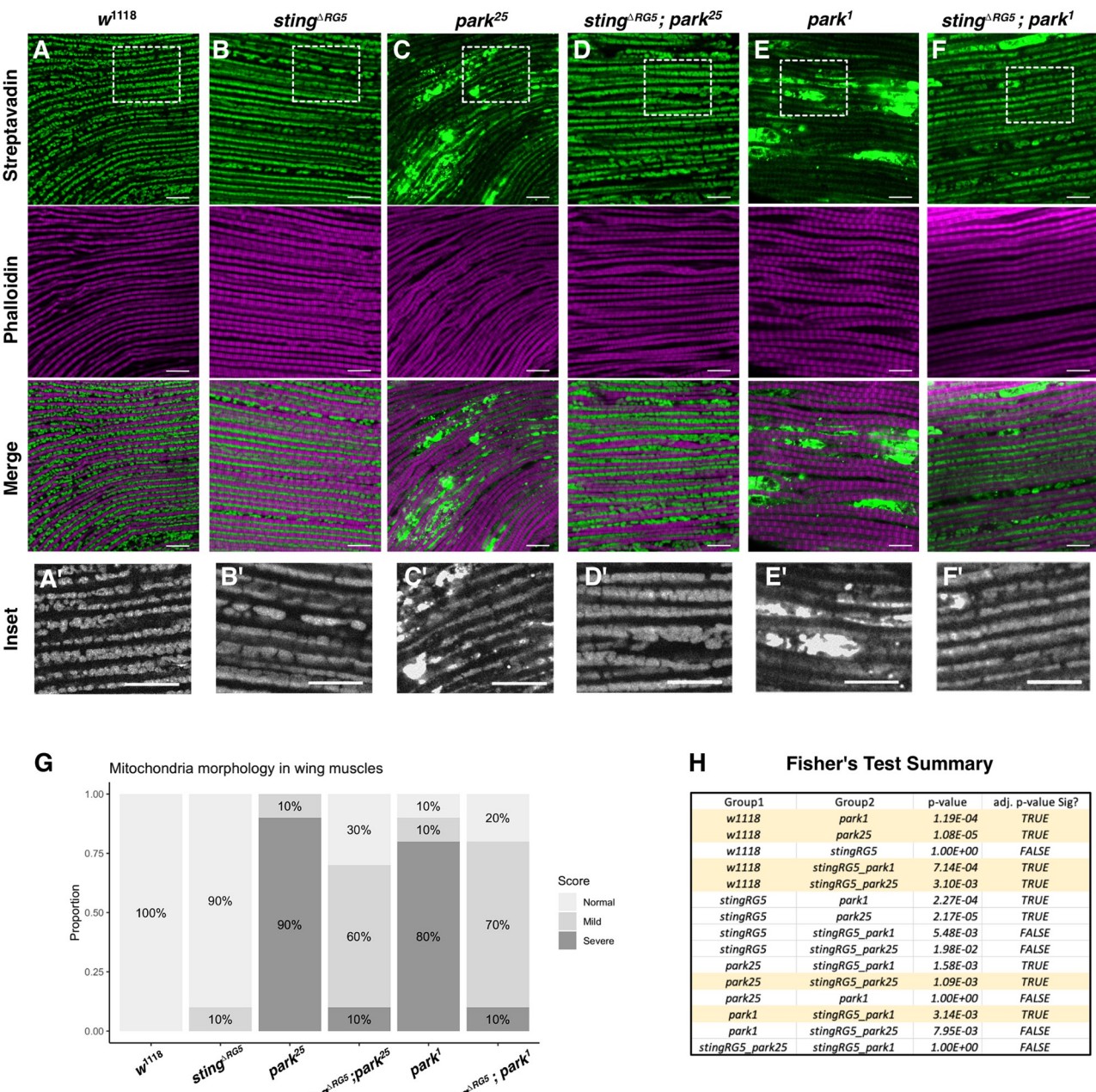

**Fig 2. Mitochondrial defects in *parkin*<sup>-/-</sup> flight muscles are suppressed by mutation of *sting*.** Representative micrographs from indirect flight muscle tissue in *w*<sup>1118</sup> (**A**) and *sting*<sup>ΔRG5</sup> (**B**) thoraces, and in flies homozygous for either of the *parkin*-null alleles (**C** and **E**). Loss of *sting* mitigates the swollen mitochondria defects in *park*<sup>25</sup> and *park*<sup>1</sup> muscles (**D** and **F**). Staining of mitochondria was performed with AlexaFluor488-conjugated streptavidin and actin bundles were visualized with iFluor647-conjugated phalloidin. Each image is a single 1μm confocal slice. All scale bars represent 10μM. Images were linearly adjusted for brightness and contrast to avoid obscuring morphology (**A'–F'**) 2X digital zoom of the corresponding mitochondria image, indicated with white dotted box. (**G**) Quantification of mitochondria morphology with blinded analysis from 10 examples per genotype, presented randomly. Data displays the percentage of thoraces in each category for each indicated genotype. (**H**) Summary of Fisher's Exact Test's from data presented in G. Key significant comparisons are highlighted in yellow. Adj p-value < 3.33E-03 was used for cutoff.

per genotype reveals that although the mitochondria aggregation is partially suppressed, loss of *sting* does not completely restore mitochondria health (Fig 2G). These results suggest a role for STING function upstream or in parallel to the mitochondrial damage phenotypes in *parkin*$^{-/-}$ flies.

## Ubiquitous expression of STING reverts loss of *STING* but overexpression alone does not further exaggerate *parkin* mutant phenotypes

To test specificity for loss of *sting* in suppressing the *parkin*$^{-/-}$ phenotypes, flies were generated to restore expression of STING in the *sting*$^{ARG5}$; *park*$^{25}$ background. The *park*$^{25}$ allele was recombined with a pAttB-UAS-STING-V5 transgene and with the ubiquitous driver *Daughterless*-Gal4 (*Da*.Gal4). Overexpression of STING with *Da*.Gal4 in *parkin* wild-type animals had no effect on *parkin*-related thorax phenotypes or mitochondria morphology (Fig 3B and 3D). These two chromosomes were moved into the *sting*$^{ARG5}$ mutant background, and then crossed together. The progeny expressing STING-V5 in a *sting*$^{-/-}$ and *parkin*$^{-/-}$ mutant background had high penetrance of thorax indentations and bent-down wings compared with sibling flies or progeny from a control cross to *sting*$^{ARG5}$; *park*$^{25}$ with no Gal4 (Fig 3A–3C, 3G and 3H). The slightly increased proportion of bent wings and small disruptions in the mitochondria networks in the *sting*$^{ARG5}$;*park*$^{25/25}$, UAS-STING flies are potentially due to "leaky" expression of the UAS-STING allele. Additionally, *hs70*-Gal4 driven expression of STING-V5 in *park*$^{25}$ mutant flies did not affect the severity of the thorax indentations or mitochondria morphology (Fig 3B and 3F). Together, these results indicate that STING is involved in development of muscle degeneration of *parkin*$^{-/-}$ flies but increasing expression of STING is not sufficient to induce damage.

## Apoptosis is reduced in *sting; parkin* flies, whereas phosphorylated Ub is elevated

Apoptotic nuclei appear in the IFM of *parkin* mutant flies shortly following enclosing and cell death persists throughout adulthood [14,17]. To test whether loss of STING protects flies from muscle apoptosis, thoraces from flies aged one-day post-eclosion were dissected and TUNEL staining was performed to detect apoptotic nuclei (Fig 4A). A high number of TUNEL-positive nuclei were observed in the *park*$^{25}$ mutant flies (Fig 4B). Loss of *sting* significantly suppressed the number of TUNEL-positive nuclei (Fig 4A and 4B), suggesting that STING is promoting apoptosis in the *parkin* mutant flies.

To assess whether activation of the PINK1/Parkin pathway was affected in STING-null flies, western blotting for phosphorylated-Serine65 of Ubiquitin was performed (Fig 4C). No change was detected in the amount of p-S65-Ub due to loss of *sting* alone (Fig 4D). Consistent with a previous report, *park*$^{25}$ mutants display a high amount of p-S65-Ub, attributed to high basal PINK1 activity and decreased capacity to degrade ubiquitinated proteins via the proteasome or mitophagy [41]. We confirmed this result and replicated this in *park*$^{1}$ mutants as well. Deletion of *sting* in either of the *parkin* mutant backgrounds further increased the amount of p-S65-Ub (Fig 4D). Western blots on protein samples isolated from dissected thoraxes, with the gut tract removed, confirmed that a similar increase in p-S65-Ub occurs in the thoracic wing muscle of *sting*$^{ARG5}$; *park*$^{25}$ mutants (S3A and S3B Fig). As it was unclear whether the rescued mitochondrial morphology and decreased cell death contributes to the increase in the relative amount of p-Ubiquitin, we tested these samples already normalized for total protein levels for the mitochondrial respiratory Complex V subunit ATP5α (S3C Fig). ATP5α levels were slightly lowered in *pink1* and both *parkin* mutants, and deletion of *sting* slightly increased amount of ATP5α (S3C and S3D Fig). When p-Ubiquitin is normalized to the relative amount

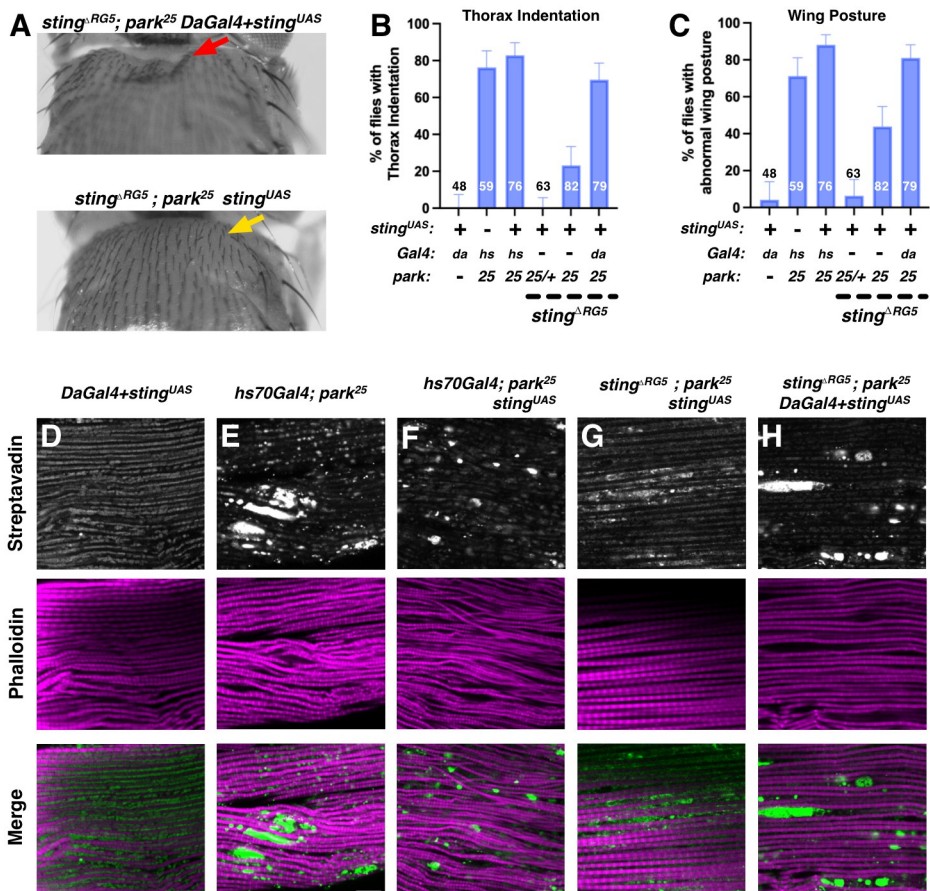

**Fig 3. Overexpression of STING reverts suppression of *park* phenotypes from deletion of *sting* without further increasing phenotype severity.** (**A**) Examples of flies from crosses testing over-expression of a UAS-STING transgene. Re-expression of STING with the ubiquitous *daughterless*-Gal4 restored the *parkin* phenotypes in an otherwise *sting*[-/-]; *park*[-/-] background. (**B & C**) Quantification of thorax indentation and wing posture phenotypes in multiple UAS-STING overexpressing flies. Overexpression of UAS_Sting with da.Gal4 did not result in wing or thorax defects in wild-type flies (harboring two copies of parkin). Note that with the *hs70*-Gal4 driver, overexpression of Sting in *sting* wildtype but *parkin* mutant flies did not increase the severity or proportions of *parkin* mutant thorax defects. Error bars represent the 95% confidence interval for the population proportion and the numbers indicate number of flies scored. (**D-H**) Representative images of mitochondria morphology in IFM samples of the indicated genotypes. Samples were imaged and examined in a blinded manner. All scale bars represent 10μM and the images were linearly adjusted for brightness and contrast to avoid obscuring morphology.

of mitochondria protein, the difference in *parkin* flies and the *sting*; *parkin* double mutant flies is less severe, although still increased, than observed in the un-normalized data (Figs 4D and S3E). Together these results demonstrate that deletion of STING does not suppress phosphorylation of Ubiquitin at Ser65, and that this Pink1-mediated pathway remains activated. We also assessed the levels of p62 (dm: *ref(2)p*, hs: *SQSTM*), a major autophagy receptor in flies, which has been implicated in regulation of *pink1/parkin*-dependent mitophagy [42] and overexpression of p62 suppresses mitochondria dysfunction in muscles associated with aging [43]. Western blotting against p62 reveals an increase in p62 in the rescued *sting*[ΔRG5]; *park*[25] animals (S3F and S3G Fig), coinciding with the observed increase in pSer65-Ub and prevention of mitochondria turnover.

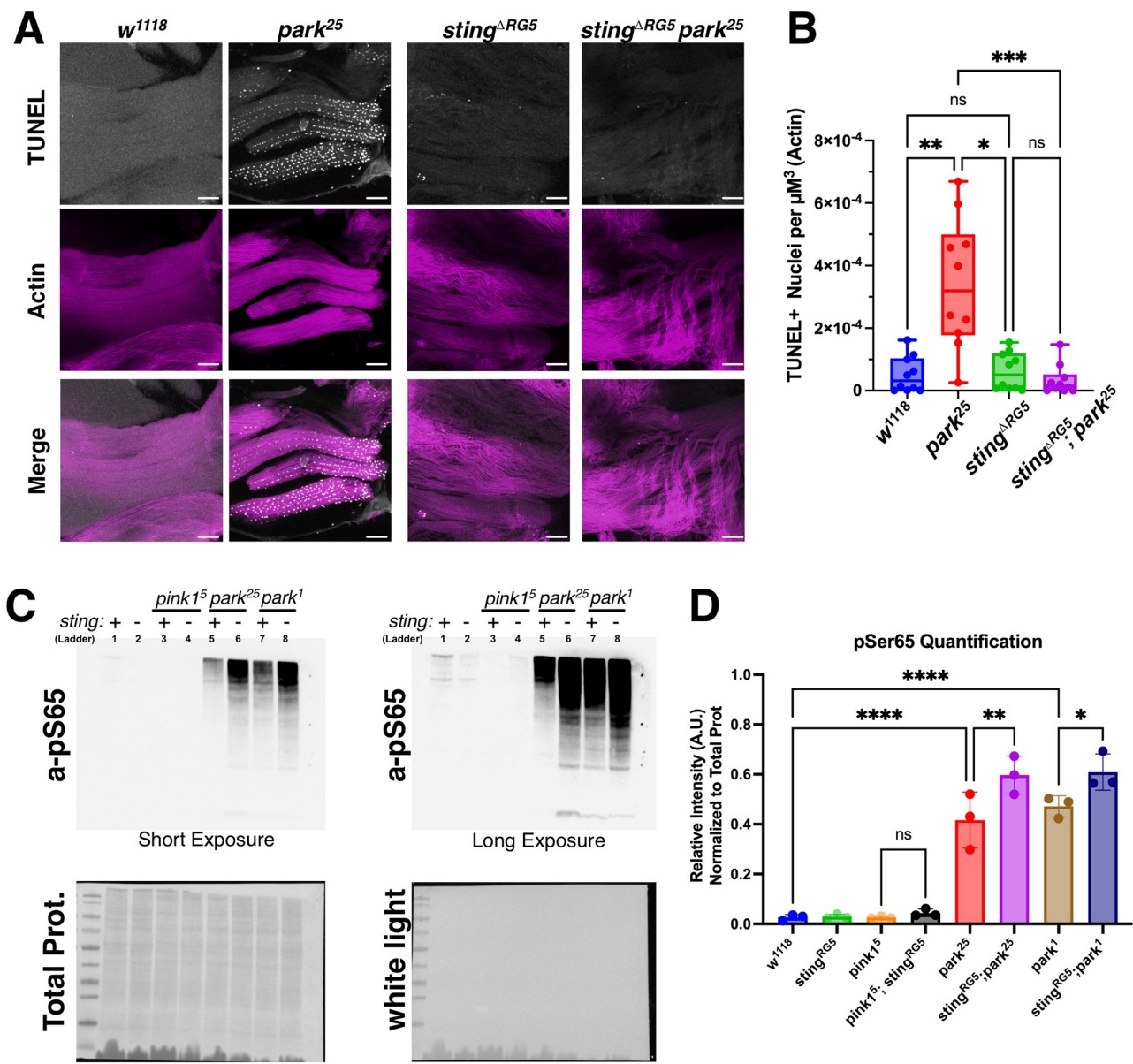

**Fig 4. Loss of STING suppresses cell death without preventing Ubiquitin phosphorylation.** (**A**) Example images of TUNEL staining on thoracic muscle from the indicated genotypes. Images are max projection stacks of 10 slices at 1μm step size. Scale bars represent 50μm. (**B**) Quantification of relative number of TUNEL-Positive Nuclei. Data is graphed as number of positive nuclei per μm³ volume of muscle quantified from phalloidin staining. N = 3 biological replicas, Significance was determined based on the results of the Kruskal-Wallis test, with Dunn's multiple comparisons test. (**C**) Normalized protein lysates from 5 adults flies of the indicated genotypes were subjected to SDS-PAGE followed by Western blotting for pSer65 Ubiquitin. (**D**) Quantification of lanes from pSer65-Ub western blots. N = 3 biological replicas, Significance was determined based on the results of an 1-way ANOVA, followed by Bonferroni's multiple comparison's testing.

## Canonical STING signaling is not activated in young *parkin* flies

STING is reported to act upstream of the NF-κB transcription factor Relish in Drosophila and regulate both anti-bacterial and anti-viral responsive genes [31,44–46]. We assayed STING-dependent response genes in *parkin* mutants and control flies and observed no aberrant activation of the STING-regulated anti-viral genes *srg2* (CG42825) and *srg3* (CG33926) in *parkin*⁻ᐟ⁻

flies with RT-qPCR (S4A–S4D Fig). To test the hypothesis that decreased *relish* signaling is involved in the rescue of the *parkin* fly phenotype by deletion of *sting*, we generated fly lines with the *park$^{25}$* and *rel$^{E20}$* null deletions recombined [47]. This combination of homozygous mutants results in lethality as among greater than 200 flies collected from three independent recombined lines, no homozygous *park$^{25}$*, *rel$^{E20}$* flies were observed (S4E Fig). We hypothesize that, since *park$^{-/-}$* flies are hypersensitive to bacteria propagation [48,49], the combination of defects from loss of *rel* result in synthetic lethality, possibly distinct from the role of *sting*-mediated immune responses in *park$^{-/-}$* flies. Further, an allele harboring null mutations of two cGAS-Like receptors, cGLR1$^{ko}$ and cGLR2$^{ko}$ [50] failed to completely replicate the loss of *sting* with regards to the *parkin$^1$* phenotypes (S4F Fig). In the *cGLR1$^{ko}$*, *cGLR2$^{ko}$*; *parkin$^1$* flies, only a minor decrease of the thorax phenotype penetrance was observed, and there was no effect on the severity of the wing posture defects. We then assayed levels of mtDNA, a putative, yet untested, cGLR-activating signal. From total column-purified DNA samples, the mtDNA copy number (normalized to nuclear DNA levels) was significantly lowered in *parkin* mutants, compared to the *w$^{1118}$* background controls. Loss of *sting* returned these mtDNA levels to approximately that of wild-type (S4G Fig). This supports the hypothesis that the disruption of mitochondrial homeostasis in *parkin* mutants is suppressed by deletion of *sting* (see also Fig 2). These findings and the evidence that loss of STING prevents the mitochondria morphology defects suggest that STING's role in the *parkin$^{-/-}$* flies may be separate from the reported function in anti-viral innate immunity.

## Transcriptomes of *sting$^{ΔRG5}$;park$^{25}$* flies implicate additional stress-response and innate immune pathways

The unexpected result of the improved mitochondria morphology and the lack of an increase in STING-regulated expression of two anti-viral genes in the *park* mutant suggests more complex models for the suppression of *park* phenotypes when *sting* is mutated. Therefore, we performed RNA-sequencing to compare the transcriptomes of the *sting$^{ΔRG5}$*, *park$^{25}$*, and *sting$^{ΔRG5}$; park$^{25}$* mutant flies, using the shared background stock *w$^{1118}$* as our wild-type control. Samples of total RNA from ten age-matched male flies (4–5 days post-eclosion) were prepared for RNA-sequencing. Following sequencing and preliminary analysis for quality control, at least two independent replicas per group were used for differential gene expression and gene set enrichment analysis. We verified that the expression levels of the STING-regulated genes *srg2* (CG42825) and *srg3* (CG33926) were significantly lower in both *sting* mutant groups and found that, in contrast, anti-viral *srg1* is slightly increased in *park$^{25}$* mutant flies and not in *sting$^{ΔRG5}$; park$^{25}$* mutant flies (S4H Fig). Some of IMD/Relish mediated antimicrobial peptides, previously linked to STING activity following infection with *Listeria monocytogenes* [31], were shown to be elevated in the *park* mutants, which matches prior reports of AMP activity in *parkin* flies (S4H Fig). The most significantly upregulated GO term category in the *park* vs wildtype transcriptome comparison is antibacterial humoral response (Fig 5A). Thus, although it remains unclear if STING-mediated transcriptional responses are involved in the *parkin* fly phenotypes, if so, it would appear that antibacterial responses would be more important than anti-viral responses.

Compared with the wild-type controls, *parkin* mutant samples have consistently lower expression of genes involved in mitochondrial respiration (Fig 5A, left panel), and expression of these genes was rescued in the double mutants (Fig 5A, right panel). One significantly enriched gene set in the double mutant flies is genes involved in glutathione metabolic processes (GO:0006749, KEGG: N00904) (Figs 5A and 5B and S5). Compared to wild-type controls, *parkin* flies also have increased heat-responsive and humoral immune-response genes

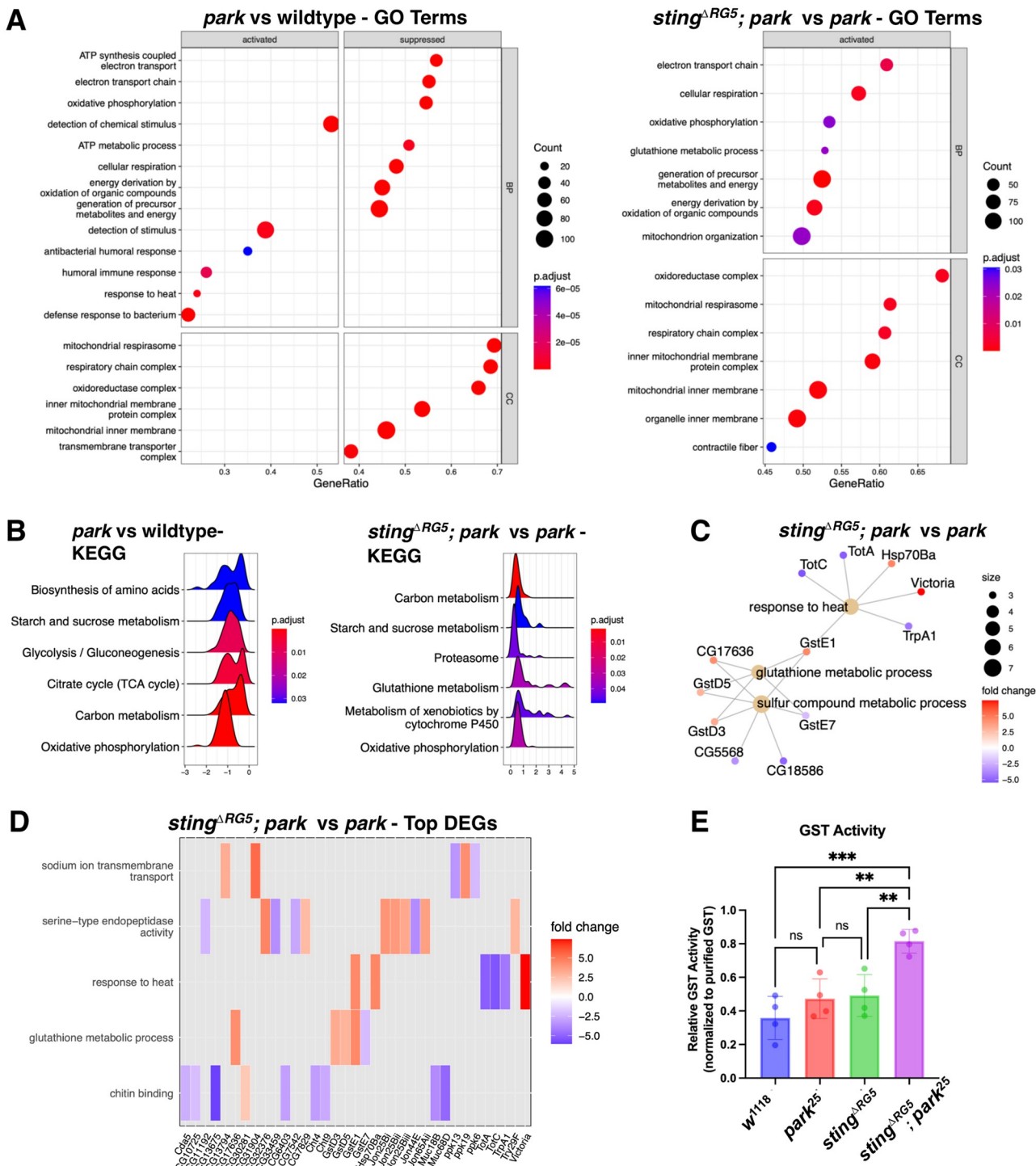

**Fig 5. Transcriptomic profiling reveals a role for pro-survival and stress responsive genes in suppressing *parkin* phenotypes.** (**A**) Gene set enrichment analysis (GSEA) for RNA-sequencing results, comparing either (left) *parkin* to wild-type control, or (right) the *sting^{ΔRG5}; park^{25}* double knockout to *parkin* mutants. Enriched Gene Ontology (GO) terms are displayed as a dot plot, separating biological processes (bp) from cellular component sets (CC). Count indicates number of genes in the GO set, and p-values represent the adjusted p-value using the BH method. Note that in the double knockout vs *parkin* set the only significantly enriched sets were activated (higher in the double mutant). (**B**). Ridgeplots of the results from GSEA with the KEGG classifications network. Graphs indicate the distribution of expression levels for significantly enriched KEGG sets for analysis of *parkin* compared to wild-type samples, and *sting^{ΔRG5}; park^{25}* to *parkin* mutants. (**C**) Gene concept network after overrepresentation analysis (ORA) of the top differentially expressed genes between the *sting^{ΔRG5}; park^{25}* and *parkin* sets. Significance was determined with an adjusted FDR cutoff of 0.05 and Log2FC cutoff of 2. (**D**) Heatmap plot indicating expression levels of the top differentially expressed genes between the *sting^{ΔRG5}; park^{25}* double

knockout and *parkin* mutants, graphed by gene and gene family set. (**E**) Activity assays for GST conjugation from thorax lysates of 10 flies of the specified genotype. Data is shown as percent activity compared to the positive purified GST control. Results from four biological replicas. Significance was determined based on the results of 1-way ANOVA, followed by Bonferroni's multiple comparison's test.

(Fig 5A, left panel). Examination of the highest enriched genes (Figs 5C and 5D and S5C) suggests that these changes come in part from higher expression of Turandot genes, a family of heat-response and oxidative stress-induced genes [51]. TotA and TotC are highly enriched in the sick *parkin* mutant flies, are significantly decreased in the double mutants, and lowest expressed in the two control groups ($w^{1118}$ and *sting*$^{-/-}$) (Figs 5D, S5C and S5D).

Due to a previously established connection to *parkin* phenotypes, we hypothesized that increased GST activity could lessen the burden of toxic species in the *sting*$^{ΔRG5}$; *park*$^{25}$ double mutant animals. The relative activity of GST enzymes in fly thorax protein extracts was investigated using a GST enzymatic assay. Loss of either *parkin* or *sting* had a slight nonsignificant increase in GST activity, however loss of both genes resulted in an increase of approximately 2-fold activity compared to the wild-type samples (Fig 5E). Given this relatively small increase in GST activity in the adult flies there may yet be additional undiscovered factors improving the health of these animals.

## Discussion

Together, our findings support a non-canonical role for Drosophila STING in the pathogenesis of mitochondria dysfunction in *parkin*$^{-/-}$ flies. Based on rescued mitochondria health and suppression of apoptosis, we propose that Drosophila STING is not responding solely to the presence of mitochondria-derived damage signals in the *parkin* mutants. These findings suggests that instead in flies there is an indirect role for STING or additional STING-induced genes in propagating upstream mitochondria-damage-induced signaling or indirectly promoting apoptosis. Additionally, there may also be a general dysregulation of autophagy or intraorganellar signaling in the STING-null mutants, as recent studies show that STING modulates autophagy [45,52,53] and lipid dependent starvation responses [54]. Boosting of autophagy through expression of ATG1 has also been shown to prevent mitochondria aggregation and rescue phenotypes of the *parkin*$^{-/-}$ flies [38]. We have identified a significant increase in p62 levels in the double mutant animals, suggesting either a block in autophagic turnover, or increased expression of p62. Increased amounts of p62 promotes longevity in flies [42,43] and promotes pro-survival NRF2 (CnC in flies) activity through an inhibitory interaction with the NRF2 regulator KEAP1 [55,56].

Previous work demonstrates that Drosophila STING's function in innate immunity requires activation of the IMD (immune deficiency) pathway leading to increased Relish (NF-κB) signaling and this activation is partly dependent on the Drosophila IKKβ homologue [44,46]. Supporting this requirement for IKK signaling, the Drosophila IKKε homologue has been demonstrated to interact genetically with *parkin* mutations, as loss of IKKε suppressed the *parkin* wing and thorax phenotypes [57]. IMD/Rel-induced AMPs were previously upregulated in a transcriptomic study of *parkin*$^{-/-}$ mutants [27] and such AMPs are reported to promote neurodegeneration in aging animals [58]. A possible mechanism for our observations is that minor damage to mitochondria could signal through STING to induce antimicrobial gene expression that feeds back on mitochondria to cause unmitigable damage when Parkin is absent. The RNA-Seq analysis revealed a slightly increased expression of AMPs that are regulated by the canonical IMD/Rel pathway- including Attacin (*AttA*) and Diptericin (*DptA*)- or the MyD88/Toll pathway- Drosomycin (*Drs*) and Metchnikowin (*Mtk*)- in young *parkin*$^{-/-}$

flies. However, as tested with qPCR (S4A–S4D Fig), and corroborated with RNA-seq results (S4H Fig), the expression levels of two anti-viral STING-regulated genes were not consistently increased in the adult *park*[-25] samples. Combination of a mutant allele lacking two cGAS-like Receptors *cGLR1* and *cGLR2* with the *parkin*[1] mutant animal has a slight, but significant suppression of the *parkin* mutant thorax indention penetrance, indicating that cGLR1/cGLR2 may be dispensable for the role of STING in the fly *parkin* phenotype. There exist additional cGAS-like-receptors and knockout of all these simultaneously in *parkin* flies would be intriguing, yet technically challenging [50,59,60]. Recent evidence suggests that *Drosophila* STING possesses additional functions independent of the canonical activation of NF-kB innate immune signaling genes, such as regulation of autophagy or metabolism related pathways [45,53,54].

RNA-sequencing results and previously published microarray data from *sting* mutants [54] supports that there are additional cellular pathways dysregulated in flies lacking *sting* besides immune-related genes. The observed increase in Glutathione S-transferase enzyme expression could convey cytoprotective antioxidant buffering to the double mutant flies. Elevated expression of Glutathione S-transferases [32,61,62], toxic metal responsive genes such as MTF-1 [63], and increased activity of the antioxidative stress KEAP1/NRF2 pathway —which regulates GST gene expression—have each been demonstrated to suppress muscle and/or climbing phenotypes in *parkin* or *pink1* mutant flies [32,61–64]. A previously published dataset (GEO accession #GSE167164) shows an upregulation in anti-toxin and anti-pesticide genes such as GstD1 and Cytochrome p450 family members in *sting*[-/-] mutants [54]. The transcriptomic analysis we performed did not reveal as strong of a change in these genes between the *sting*[ΔRG5] mutant and the wildtype control, however, we did observe a significant increase of GstE1, GstE11, and GstD2 in the *sting*[ΔRG5]; *park*[25] mutant samples. Furthermore, *sting* mutant flies were shown to have metabolic changes related to β-oxidation and lipid storage, which may influence mitochondria bioenergetics and promote antioxidant responses [54]. We propose that an increase in oxidative stress responses and GST activity could contribute to the improved outcomes of *sting*[ΔRG5]; *park*[25] animals, however there may yet be additional signaling factors during the developmental larval and pupal stages.

Additionally, it remains unknown exactly why loss of *sting* fails to rescue *pink1* at the same degree observed in *parkin* mutant alleles. We hypothesize that Parkin-independent components are contributing to the muscle degeneration in *pink1* mutants, therefore loss of *sting* fails to suppress these phenotypes completely. A recent study on *pink1* mutant flies implicate a different DNA-recognition receptor, EYA, as contributing to the severity of some neuronal and gut-based *pink1* mutant phenotypes through regulation of Relish signaling [65]. This contribution may be similar or completely independent of STING's function in *parkin* pathology. Additionally, our observed differences in the amount of Ubiquitin phosphorylation could reflect increased amounts of Pink1 activity, the Ub substrates, or a decrease in deubiquinating enzymes. The molecular details of phosphorylated-Ub regulation remains of high interest to the *pink1/parkin* field [41,66], reviewed recently in [67].

In summary, loss of *sting* in flies suppresses the severe phenotypes of *parkin* mutants, through a mechanism(s) independent of the canonical role in innate immunity signaling. The candidate pathways supported by our data includes anti-oxidative stress responses and activation of cell death pathways. These underlying changes to the transcriptional landscape in *sting*[-/-] flies necessitates further study to better understand the role of stress-responsive genes in mitigating mitochondrial and oxidative damage during fly development or disease.

## Materials and methods

### Experimental subject details

Publicly available fly stocks (details in Table 1) were acquired from Bloomington Drosophila Stock Center (BDSC, Bloomington, IN). Experimental genotypes (see S1 Table for all genotypes) were made using classical genetics, utilizing the balancer chromosomes from $w^{1118}$; $wg^{Sp-1}/CyO; MKRS/TM6b, hu$ (BDSC stock #76357) when necessary. The null $sting^{ΔRG5}$ allele was gifted from Dr. Alan Goodman, Washington State University and was previously described [31]. The $park^{25}$ allele was acquired from Dr. Alicia Pickrell, Virginia Tech University, and originally generated by Dr. Leo Pallanck, University of Washington [14]. All $park^{25}$ mutant animals were maintained as heterozygous over the *TM6b, Hu* balancer and routinely checked with PCR for presence of the deletion. A second stock of $sting^{ΔRG5}; park^{25}/TM6b, hu$ flies were gifted to us from Dr. Alexander Whitworth. Male flies from these stocks were crossed to a $w^{1118}$ background, then outcrossed for 6 further generations. After each other generation, single male flies used in crosses were checked for PCR after the cross was seeded, and only the ones carrying the $park^{25}$ allele were selected. After 7 generations, single males were crossed to the double balanced stock for maintaining the outcrossed alleles, and again, PCR was used to confirm the presence of the $park^{25}$ allele.

 *pink1[5]/FM7* female flies were outcrossed to *w*[1118] males. After the first cross, freely recombining *pink1[5]/w*[1118] females were crossed to a *FM7/y;* CyO/+ male to ensure the X-chromosome *pink1[5]* allele was recovered. From there, multiple *pink1[5]/FM7; +/CyO* flies were crossed with the *sting[RG5]* allele to generate the *pink1[5]/FM7; sting[RG5]/CyO* candidate lines then PCR and phenotyping was used to confirm the *pink1* genotype. *pink1[B9]/FM7* females were crossed with a FM7; *sting[RG5]/CyO* male for two generations to generate *the pink1[B9]/FM7; sting[RG5]/CyO* stock. All flies used in experiments were raised on vials or bottles with Jazz Mix food (Thermo Fisher Scientific, P/N AS153), reconstituted in MilliQ water and prepared per recipe instructions. For experiments, flies were raised in a 25°C incubator on a standard L:D cycle with humidity control.

### Generating new UAS-STING-V5 alleles

For generating the pUASTattB_UAS_STING insertion, the *sting* cDNA was PCR-amplified from LP14056 BDGP gold cDNA (DGRC stock #1064136, FlyBaseID: FBcl0189577, RRID: DGRC_1064136) and subcloned into a modified pUAST-attB (DGRC Stock #1419, RRID: DGRC_1419) with NEBuilder HiFi DNA Assembly Mix (New England Biolabs, P/N E2621). The C-terminal V5 sequence had been inserted with two annealed oligos ligated into the XhoI and XbaI sites on pUASTattB. After verifying with sequencing, plasmids were sent to BestGene (BestGene Inc. Chino Hills, CA) for injection services using 62E1 attP landing site flies, BDSC stock #9748. The Phi31C source was removed, and the mini-white positive progeny were used to established balanced lines. The UAS-STING allele was recombined with the $park^{25}$ allele, and then assayed with PCR genotyping and verified with outcross and observation of the *parkin* homozygous phenotype.

### Wing muscle and thorax phenotyping

Flies were anesthetized with $CO_2$ and thoraces were examined under a dissecting microscope. Flies were scored for thorax shape and wing posture within the first 5 minutes of anesthesia. Blinding of genotypes to observer was performed when practical, including all of the initial assays involving the key genotypes in Fig 1.

**Table 1. Materials and Critical Resources.**

| REAGENT or RESOURCE | SOURCE | IDENTIFIER |
|---|---|---|
| **Antibodies** | | |
| Phospho-Ubiquitin (Ser65) (E2J6T) Rabbit monoclonal Ab | Cell Signaling Technology | Cat #62802; RRID:AB_2799632 |
| Anti-alpha-Tubulin Mouse Monoclonal Ab | Sigma-Aldrich | Cat #T6074; RRID:AB_477582 |
| V5-Tag (D3H8Q) Rabbit monoclonal Antibody | Cell Signaling Technology | Cat #13202; RRID:AB_2687461 |
| Anti ATP5α Mouse monoclonal Ab [15H4C4] | Abcam | Cat # ab14748; RRID: AB_301447 |
| Anti p62/ref(2)p- Rabbit polyclonal | Gift from H. Kramer, UTSW. | N/A |
| IRDye 800CW Goat anti-Rabbit IgG | LI-COR Biosciences | Cat #926–32211; RRID: AB_621843 |
| IRDye 680RD Goat anti-Mouse IgG | LI-COR Biosciences | Cat #926–68070; RRID: AB_10956588 |
| Amersham ECL Donkey anti-Rabbit IgG, HRP-linked whole Ab | Cytiva | Cat #NA934; RRID:AB_772206 |
| Amersham ECL Sheep anti-Mouse IgG, HRP-linked whole Ab | Cytiva | Cat #NA931; RRID:AB_772210 |
| **Bacterial and virus strains** | | |
| DH5alpha Competent Cells- *Escherichia coli* | New England Biosciences | Cat #C2987H |
| **Chemicals, peptides, and recombinant proteins** | | |
| Alexa Fluor 488-Streptavidin | Jackson ImmunoResearch | Cat # 016-540-084 |
| Phalloidin-iFluor 647 Conjugate | Cayman Chemical Company | Cat #20555 |
| 16% Paraformaldehyde Aqueous | Electron Microscopy Sciences | Cat #15710 |
| Prolong Gold Antifade Solution | Thermo Fisher Scientific | Cat #P36930 |
| Ponceau Red Total Protein Stain | Thermo Fisher Scientific | Cat #A40000279 |
| Amersham ECL | Cytiva | Cat #RPN2232 |
| SuperSignal Femto ECL | Thermo Scientific | Cat #34095 |
| Tri Reagant | Zymo Research | Cat #R2050-1-200 |
| cOmplete Protease Inhibitor cocktail | Sigma-Aldrich | Cat #4693159001 |
| PageRuler Plus Prestained 10 250kDa Protein Ladder | Thermo Fisher Scientific | Cat #26620 |
| **Commercial assays** | | |
| Glutathione S- transferase Assay Kit | Cayman Chemical Company | Cat No. 703302 |
| DirectZol | Zymo Research | Cat No. R2050 |
| High Capacity Reverse Transcription Kit | Thermo Fisher Scientific | Cat #4368814 |
| Applied Biosystems PowerUp SYBR Green Master Mix | Fisher Scientific | Cat #A25776 |
| Itaq Universal Probes Supermix | Bio-Rad | Cat #1725130 |
| Quick-DNA Miniprep Plus Kit | Zymo Research | Cat #D4068 |
| Q5 High-Fidelity DNA Polymerase kit | New England Biosciences | Cat #E0555S |
| NEBuilder HiFi DNA Assembly Master Mix | New England Biosciences | Cat #E2621 |
| TmT-Red Cell Death Detection Kit | Roche (Sigma-Aldrich) | Cat #12156792910 |
| **Experimental models: Organisms/strains** | | |
| *D. melanogaster*: w[1118] | Gift from Dr. Alicia Pickrell | FBID: FBal0018186 |
| *D. melanogaster*: w[1118]; sting[ΔRG5] | Gift from Dr. Alan Goodman; Martin et. al. 2018 [30] | FBal0340353 |
| *D. melanogaster*: w[1118]; sting[ΔRG5]; TM6B/TM3 Ser | This Study | N/A |
| *D. melanogaster*: w[1118]; P{GAL4-hsp70}; park[25]/TM6B | Gift from Dr. Alicia Pickrell; Sharraf et.al 2019 [53]. *park*25 originally reported in Green et.al. 2003 [13] | FBID: FBal0146938 |
| *D. melanogaster*: w[1118];; park[25]/TM6B | This paper | N/A |
| *D. melanogaster*: w[1118]; sting[ΔRG5]; park[25]/TM6B | This paper | N/A |
| *D. melanogaster*: w[1118]; sting[ΔRG5]; park[1]/TM6B | This paper | N/A |

*(Continued)*

**Table 1.** (Continued)

| REAGENT or RESOURCE | SOURCE | IDENTIFIER |
|---|---|---|
| D. melanogaster: w[1118];; park[1]/TM3 | Bloomington Drosophila Stock Center, Cha et.al., 2005 [14] | RRID: BDSC_34747 FBID: FBal0189571 |
| D. melanogaster: y[1] w[1118];; P{mW+ UAS-Sting[WT]} | This Study, injections from BestGene, Camarillo, CA | AttP Source: RRID: BDSC_9748 |
| D. melanogaster: w[1118]; sting[ΔRG5]; P{mW+ UAS-Sting[WT]} | This paper | N/A |
| D. melanogaster: w[1118]; sting[ΔRG5]; park[25] P{mW+ UAS-Sting[WT]}/ TM6B | This paper | N/A |
| D. melanogaster: w*;; P{GAL4-da.G32}UH1 Sb1/TM6B | Bloomington Drosophila Stock Center | RRID: BDSC_55851; FBID: FBst0055851; |
| D. melanogaster: w[1118]; sting[ΔRG5]; P{GAL4-da.G32}UH1 Sb1 | This paper | N/A |
| D. melanogaster: w[1118]; sting[ΔRG5]; park[25] P{GAL4-da.G32}UH/ TM6B | This paper | N/A |
| D. melanogaster: w*; sting[ΔRG5]/CyO; park[25]/TM6B | Gift from Dr. Alexander Whitworth; Lee et.al. 2020 [28] | N/A |
| D. melanogaster: relish[E20] | Bloomington Drosophila Stock Center | RRID:BDSC_55714 |
| D. melanogaster: w*; CyO/Kr-lf; relish[E20] park25/ TM6B | This paper | N/A |
| D. melanogaster: cGLR1, cGLR2 double-KO | Gift from Jean-Luc-Imler.; Holleufer A, et.al. 2021 | N/A |
| D. melanogaster: cGLR1+2 KO/CyO; park1/TM6B | This Paper | N/A |
| D. melanogaster: w[1118]; wg[Sp-1]/CyO; MKRS/TM6B, Tb[1] | Bloomington Drosophila Stock Center | RRID: BDSC_76357 |
| D. melanogaster: w* pink1[5]/FM7i, P{ActGFP}JMR3 | Bloomington Drosophila Stock Center | RRID: BDSC_9748; FBID: FBal0196293 |
| D. melanogaster: w* pink1[5]/FM7i; sting[ΔRG5] | This paper | FBID: FBal0196293 |
| D. melanogaster: w* pink1[B9]/FM7 | Gift from Ed Giniger, NIH, Bethesda. Originally from Bloomington Drosophila Stock Center | RRID: BDSC_34749 FBID: FBal0193144 |
| D. melanogaster: w* pink1[B9]/FM7 sting[ΔRG5] | This paper | FBID: FBal0193144 |
| **Oligonucleotides** | | |
| Park25FWD: GATTGGCAACACTGAAGC | Greene et.al. 2005 [26] | N/A |
| Park25REV: CTTTACCATCCCCCAATCAA | Greene et.al. 2005 [26] | N/A |
| StingGeno FWD: ATTGTAGCCACCGTGTT | This paper | N/A |
| StingGeno REV: ACGTAATCTTTGGAATCGTT | This paper | N/A |
| StingcDNA-UAS: ACTCTGAATAGGGAATTGGGAATTCATGGCAATCGCTAGCAACG | This paper | N/A |
| StingcDNA-UAS REV: GGGGATGGGCTTGCCGGTACCGTTGGAAATTTCGTCAATAGTTTTGGTTTTGTTT | This paper | N/A |
| SRG2(CG42825)qPCR_FWD: GCGTTTTGGCCCTTATTATG | Goto, A. et.al. 2018 [39] | N/A |
| SRG2(CG42825)qPCR_REV: CTTTTGTAGCCGACGCAGTG | Goto, A. et.al. 2018 [39] | N/A |
| SRG3(CG33926)qPCR_FWD: GCGACCGTCATTGGATTGG | Goto, A. et.al. 2018 [39] | N/A |
| SRG3(CG33926)qPCR_REV: TGATGGTCCCGTTGATAGCC | Goto, A. et.al. 2018 [39] | N/A |
| Sting_qPCR_FWD: CCTGATTGTGGGATTCCTTCTC | This paper | N/A |
| Sting_qPCR_REV: CATATCCAGTAGAGCGGCATTT | This paper | N/A |
| Parkin_qPCR_FWD: CACTCGTTCATCGAGGAGATTC | This paper | N/A |
| Parkin_qPCR_REV: ACCTGCCTGTAGGACATACT | This paper | N/A |
| RpL32_qPCR_Fwd: ATGCTAAGCTGTCGCACAAA | Martin et. al. 2018 [30] | N/A |
| RpL32_qPCR_Rev: GTTCGATCCGTAACCGATGT | Martin et. al. 2018 [30] | N/A |
| mtDNA(CoI)FWD: 5′-TTCTACCTCCTGCTCTTTCTTTAC | Andreazza et.al. 2019 [68] | N/A |
| mtDNA(CoI)REV: 5′-CAGCGGATAGAGGTGGATAAA | Andreazza et.al. 2019 [68] | N/A |
| mtDNA(CoI)probe: 5′-FAM-AATGGAGCTGGGACAGGATGAACT-BHQ | Andreazza et.al. 2019 [68] | N/A |
| RpL32FWD: 5′-CACCGGAAACTCAATGGATACT | Andreazza et.al. 2019 [68] | N/A |

*(Continued)*

**Table 1.** (Continued)

| REAGENT or RESOURCE | SOURCE | IDENTIFIER |
|---|---|---|
| RpL32REV: 5′-CACACAAGGTGTCCCACTAAT | Andreazza et.al. 2019 [68] | N/A |
| RpL32probe: 5′-HEX-CCAAGAAGCTAGCCCAACCTGGTT-BHQ | Andreazza et.al. 2019 [68] | N/A |
| **Recombinant DNA** | | |
| pUAST-attB_dSTING_V5 | This paper | N/A |
| STING cDNA- LP14056 BDGP gold cDNA | N/A | RRID: DGRC_1064136; FBID: FBcl0189577 |
| pUAST-attB | N/A | RRID: DGRC_1419 |
| **Software and algorithms** | | |
| Graphpad Prism v.9.4.0 | GraphPad | https://www.graphpad.com/updates |
| FIJI (FIJI is Just ImageJ) | Schindelin, J.et.al. 2012[64] | https://imagej.net/software/fiji/ |
| CFX Manager Software | Bio-Rad Instruments | https://www.bio-rad.com/en-us/product/previous-qpcr-software-releases?ID=OO2BB34VY |
| R 4.2.2 | R Project | https://cran-archive.r-project.org/bin/ |
| RStudio v. 2022.07.2 | RStudio (Posit) | https://posit.co/products/open-source/rstudio/ |
| clusterProfiler v 4.6.0 | Yu, G et.al. 2012 [62] | https://doi.org/doi:10.18129/B9.bioc.clusterProfiler |
| D. melanogaster gene ontology categories | FlyBase | Version |
| **Other** | | |
| Jazz Mix Fly Food | Thermo Fisher Scientific | Cat #AS153 |
| 0.45 μm Nitrocellulose Membrane | BioRad | Cat #1620115 |
| SurePAGE 4–12% Bis-Tris gels | GenScript | Cat #M00653 |

## Geotaxis assays

Male flies were collected at 0-1d post-eclosion and aged 5–7 days before testing. For testing, 15 to 20 flies were added to empty 10cm vials, labeled randomly, and a key was generated to preserve identity of tested stocks. The vials were placed in a plastic holder and the flies were manually tapped to the bottom of the vials. The flies were recorded for 20–30 seconds post disruption. Videos were scored using ImageJ to mark and annotate individual flies. Climbing Index was calculated as the percentage of flies in a vial that climbed greater than 6cm of the vial during the observed 20 seconds post disruption. Five independent trials, each with three technical repeats, were performed for a total of at least 75 total flies per genotype.

## Immunohistochemistry

Flies (3–5 days old) were anesthetized with $CO_2$ and thoraces were dissected away from the head and abdomen in cold phosphate buffered saline (PBS). The hemithoraces were bisected along the median plane, using a pair of microscissors (Fine Science Tools P/N 15006–09). Hemithoraces were fixing in 4% paraformaldehyde (Electron Microscopy Sciences P/N 15710) for 20 minutes at room temperature. After fixation, tissues were washed twice in PBS and then incubated twice for 10 minutes each in PBS with 0.1% TritonX (PBST). Tissues were then blocked in 5% goat serum diluted in PBST for 30 minutes, then incubated in AlexaFluor488--conjugated streptavadin (Jackson ImmunoResearch P/N 016-540-084) and

iFluor647-conjugated phalloidin (Cayman Chemical Company P/N 20555) overnight at 4˚C on a rotator. Samples were then washed three times with PBST and once with PBS. Thoraces and separated muscle pieces were then mounted directly on a 1.5 coverslip in Prolong Gold AntiFade (Thermo Fisher Scientific P/N P36930) media. Images of mitochondria morphology were acquired on a Zeiss LSM 880 Airyscan confocal with a 63X/1.4 objective Plan-Apochromat (Carl Zeiss) at 2X digital zoom and a 34-channel GAsP detector. Airyscan processing was performed in ZEN Black software (Zeiss). Images were analyzed in ImageJ and adjusted linearly for contrast and brightness.

## Western blotting

For each genotype, 5 flies (3–5 days old) were anesthetized with $CO_2$ and thoraces were isolated. Thoraces were homogenized in RIPA buffer supplemented with cOmplete EDTA-free Protease Inhibitor (Sigma-Aldrich P/N 04693159001) and PhosSTOP- phosphatase inhibitor tablet (Roche P/N 04906845001), and then samples were incubated for 10 minutes on ice. Samples were centrifuged at 10,000xG to remove tissue remains. Protein levels were quantified using the Pierce BCA protein assay kit (Thermo Fisher Scientific P/N 23228). The protein samples were normalized and then reduced by adding Lithium Dodecyl Sulfate sample buffer (GenScript P/N M00676-250) and 0.1M DTT then heating for 5 minutes at 99˚C. Protein samples were separated on 4–12% SurePAGE, Bis-Tris gels (GenScript P/N M00654) and transferred to .45µM nitrocellulose membrane (BioRad P/N 1620115). Transfer efficiency and total protein amount was visualized using Ponceau S Staining Solution (Thermo Fisher Scientific P/N A40000279). Total protein images for each blot were acquired with ChemiDoc Imaging System (Bio-Rad Laboratories). Membranes were blocked for one hour with 3% milk or with 3% bovine serum albumin (Fisher Scientific P/N BP1600) (for pS65-Ub) in Tris-Buffered Saline with 0.1% Tween-20 (TBST), and probed overnight with the indicated antibody: anti-pUb(Ser65) (1:1000, rabbit polyclonal CST P/N 62802S), anti-a-Tubulin (1:4000, mouse mAb clone B-1-5-2, Millipore Sigma, P/N T5168), anti p62/ref(2)p (purified rabbit polyclonal Ab, a gift from Dr. Helmut Kramer, UT Southwestern Medical Center), ATP5α (1:4000, mouse mAb- Abcam P/N: ab14748) or V5-Tag D3H8Q (1:1000, Rabbit mAb Cell Signaling Technologies P/N 13202). After overnight incubation, membranes were washed 3 times with TBST and incubated for 1 hour with appropriate secondary antibodies diluted 1:10,000 in TBST+3% milk: HRP-coupled Donkey-anti-Rabbit or Sheep-anti-mouse (GE Healthcare Life Sciences), Goat Anti-Rabbit IgG IRDye 800CW-Conjugated (LI-COR Biosciences P/N 926–32211), or Goat Anti-Mouse IgG Antibody IRDye 680RD-Conjugated (LI-COR Biosciences P/N 926–68070). Blots were washed three times with TBST before imaging. For p65Ser-Ub detection, HRP-conjugated secondaries were incubated with SuperSignal Femto ECL (Thermo Scientific P/N 34095) for 3 minutes and imaged with ChemiDoc Imaging System (Bio-Rad Laboratories). All other primary antibodies were visualized with HRP-conjugated secondaries and incubation with AMersham ECL (Cytiva, P/N RPN2232). IR-conjugated secondaries were incubated simultaneously for one hour, and the blots were visualized on a LICOR Odyssey Fc multichannel imager after three washes with TBST. All images were processed, adjusted linearly for brightness and contrast, and analyzed for lane densitometry in ImageJ. Quantifications for pSer65-Ub, ATP5α, and p62/ref(2)p were first normalized to the intensity of the lane's total stain. Concerning ATP5α: differences in optimal exposure times between replicas led to the necessity to represent data as *percent of wild-type control* in each experiment replica (S3D Fig). pSer65 Ub quantifications were divided by the mean relative amount of ATP5α for each genotype to approximate the amount of pSer65 per mitochondria (S3E Fig).

## RNA isolations and RT-qPCR

RNA from 5 male flies were isolated using the Direct-zol RNA Miniprep kit (Zymo Research P/N R2050). Briefly, samples were homogenized in 300μL of Tri Reagant (Zymo Research P/N R2050-1-200) and then processed using the Zymo instructions for tissue samples. On-column DNAse treatments were performed before eluting the samples in 50μL of DPEC treated RNAse-free $H_2O$, according to the Zymo Direct-zol kit protocol (DNaseI supplied with Direct-Zol kit). Sample were tested for quantity and purity with Nanodrop. 500ng of RNA samples were used for reverse transcriptase reactions, using the High-Capacity cDNA Reverse Transcription kit (Thermo Fisher Scientific P/N 4368814). cDNA samples were diluted 1:5 before using in qPCR reactions. qPCR was performed using indicated qPCR primers (primer sequences can be found in Table 1) with PowerUp SybrGreen Master Mix (Applied Biosystems P/N A25742) using a BioRad CFX384 Touch Real-Time PCR Thermocycler. Raw data was exported from BioRad Manager then analyzed using the ddCT method with Excel. CT values were normalized to the housekeeping gene *rpl32* (also referred to as *rp49*) and then normalized to the wild-type control sample. Data from two to three independent biological replicas with three technical replicas per sample are presented.

## mtDNA copy number assays

Total DNA was extracted from 10 male flies with the Quick DNA Miniprep Plus kit from Zymogen, according to the provided instructions. Quantification of mtDNA was performed using a multiplex TaqMan assays using validated probes against the mitochondrial gene mt: CoI and the nuclear-encoded gene rpL32 for reference [68]. Approx. 7ng of template DNA was used for each reaction. Primer details can be found in Table 1. qPCR reactions were performed on the BioRad 384CX system according to information provided for iTaqMan Supermix (BioRad P/N 1725130) with annealing temps at 60˚C. Data was analyzed using the following method in Microsoft Excel: 1. mtDNA and nucDNA CT values were averaged from triplicate reactions. 2. Mitochondrial DNA content was normalized to nuclear DNA in each sample using the following equations: $\Delta CT = (nucDNA\ CT – mtDNA\ CT)$ then relative mitochondrial DNA content = $2 \times 2^{\wedge}\Delta CT$. Replica biological samples were collected and isolated on separate days. Technical replicas were performed in each qPCR reaction run. Data is presented relative to the average wild-type (w1118) mitochondrial copy number for each biological set. For statistical analysis, a 1-way ANOVA and multiple comparison testing between each of the experimental genotypes.

## GST assays

Age matched flies were collected and raised 4–5 days under standard conditions. For the assay, thoraces were dissected from ten flies per genotype/treatment/per replica were collected and immediately put on ice. The thorax samples were homogenized in GST assay sample buffer (100mM buffered potassium phosphate solution, pH 7.0, with 2mM EDTA). Samples were centrifuged at 10,000 x g for 15 minutes at 4˚C and supernatants were assayed for protein concentration using a Pierce BCA protein assay kit (Thermo Fisher Scientific P/N 23228). Samples were normalized and diluted to a protein concentration of 2μg/μL, and Glutathione S-transferase activity was measured using a GST Assay Kit (Caymen Chemicals p/n 703302). After initiating the reactions, $A_{340}$ was measured every minute for ten minutes. Rates of change were calculated from the plots of $A_{340}$ vs. time, the blank well absorbance was subtracted from each, and then the activity rate ($A_{340}$/min) was converted to estimated GST activity with the

formula:

$$\text{GST Activity(nmol/min/ml)} = \Delta\text{A}_{340}/\text{min. X } 0.00503 \, \mu\text{M}^{-1}$$

The resulting estimated activities for each technical replica (3 per biological sample) were averaged together. For each biological replica, the provided purified GST enzyme was used as a positive control, and the resulting activity for each sample is represented as percent activity compared to the purified control. Graphed data represent normalized results from four repeated experiments.

## Apoptosis detection staining

Apoptosis assays were performed on 3–4 day old flies, using the In-Situ Cell Death Detection Kit, TMR red (Roche P/N 12156792910), according to the manufacturer's instructions. Briefly, fly thoraces were dissected and bisected in freshly prepared PBS. Hemi-thoraces were fixed for 20 minutes in 4% Paraformaldehyde in PBS, pH 7.4, freshly prepared. Tissues were washed first in PBS and then in permeabilization solution (0.1% Triton X100 in 0.1% sodium citrate, freshly prepared) for 15 minutes. Samples were incubated in the TUNEL detection solution in a humidified atmosphere for 60 min at 37°C in the dark. Tissues were then washed 3 times in PBST and blocked in 5% goat serum and 3% BSA diluted in PBST for 30 minutes, then incubated in iFluor647-conjugated phalloidin (Cayman Chemical Company P/N 20555) overnight at 4°C on a rotator. Samples were washed 3 times with PBST and once with PBS. Thoraces and separated muscle pieces were then mounted directly on a 1.5 coverslip in Prolong Gold Anti-Fade (Thermo Fisher Scientific P/N P36930) media. Images of thoraces were acquired on a Zeiss LSM 880 Airyscan confocal at 20X magnification.

Quantification was performed using an ImageJ macro. In brief, stacks of 10 confocal sliced were used for max intensity projections. For each projected image, thresholding was applied to detect the phalloidin-labeled actin The muscle area was measured and the thresholded region was saved as a R.O.I. The same R.O.I. was used to count for the number of TUNEL-positive stained nuclei. The number of nuclei was then divided by the total area, to approximate the number of nuclei per $\mu\text{m}^2$. TUNEL experiments were repeated two times, and the presented data represents biological replicas of at least 10 thoraces per genotype.

## RNA sequencing and analysis

Flies were collected upon eclosion and aged 4 days in identical conditions, no more than 20 animals per vial. RNA from 10 male flies, per genotype and replica, were isolated using the Direct-zol RNA Miniprep kit (Zymo Research P/N R2050). Briefly, samples were homogenized in 300μL of Tri Reagant (Zymo Research P/N R2050-1-200) and then processed using the kit instructions for tissue samples. On-column DNAse treatments were performed before eluting the samples in 50μL of DPEC treated RNAse-free $\text{H}_2\text{O}$.

## RNA-Seq library preparation and next generation sequencing

RNA-Seq services were provided by Zymo Research Services, using their Total-RNA-Seq protocol. RNA quality was assessed with the Agilent TapeStation System. Total RNA-Seq libraries were constructed from 100ng of total RNA. rRNA depletion was performed according to standard protocol. Libraries were prepared using the Zymo-Seq RiboFree Total RNA Library Prep Kit (Cat # R3000) according to the manufacturer's instructions. RNA-Seq libraries were sequenced on an Illumina NovaSeq to a sequencing depth of at least 30 million read pairs (150 bp paired-end sequencing) per sample.

## RNA-Seq data bioinformatics analysis

The Zymo Research RNA-Seq pipeline was originally adapted from nf-core/rnaseq pipeline v1.4.2 (https://github.com/nf-core/rnaseq). The pipelines were built using Nextflow (https://www.nextflow.io/).2). Briefly, quality control of reads was carried out using FastQC v0.11.9 (http://www.bioinformatics.babraham.ac.uk/projects/fastqc). Adapter and low-quality sequences were trimmed from raw reads using Trim Galore! v0.6.6 (https://www.bioinformatics.babraham.ac.uk/projects/trim_galore). Trimmed reads were aligned to the reference genome using STAR v2.6.1d (https://github.com/alexdobin/STAR) [69]. BAM file filtering and indexing was carried out using SAMtools v1.9 (https://github.com/samtools/samtools) [70]. RNAseq library quality control was implemented using RSeQC v4.0.0 (http://rseqc.sourceforge.net/) and QualiMap v2.2.2-dev (http://qualimap.conesalab.org/)) [71,72]. Duplicate reads were marked using Picard tools v2.23.9 (http://broadinstitute.github.io/picard/). Library complexity was estimated using Preseq v2.0.3 (https://github.com/smithlabcode/preseq). Duplication rate quality control was performed using dupRadar v1.18.0 (https://bioconductor.org/packages/dupRadar/) [73]. Reads overlapping with exons were assigned to genes using featureCounts v2.0.1 (http://bioinf.wehi.edu.au/featureCounts/). Classification of rRNA genes/exons and their reads were based on annotations and RepeatMasker rRNA tracks from UCSC genome browser when applicable. Differential gene expression analysis was completed using DESeq2 v1.28.0 (https://bioconductor.org/packages/DESeq2/) [74]. Quality control and analysis results plots were visualized using MultiQC v1.9 (https://github.com/ewels/MultiQC) [75].

Further analysis and visualizations on the processed data were performed in R and Bioconductor. ClusterProfiler v.4.6 and Enrichplot v.1.19.0.01 were used for gene set enrichment analysis (GSEA) and plotting [76,77]. Heatmaps with normalized counts of highly enriched genes (absolute value of Log2 fold change greater than 3 and adjusted p.value less than 0.005) were generated with pHeatmap v.1.0.12. For all plots, ggplot2 v.3.4 and ggrepel v.0.9.2 were used for annotation. For ClusterProfiler GSEA analysis, cutoffs were: minGSSize = 50, maxGSSize = 250, and Benjamini-Hochberg adjusted p.value $<0.05$.

## Quantification and statistical analysis

Quantitative data was recorded, transcribed, and maintained in Microsoft Excel. Data set descriptions, exploration, statistics, and graphing was performed in Graphpad Prism v.9.3. Detailed data sets and all statistical test details are provided in S1 Data File. Details including data descriptors, sample size (n), and specific statistical tests can be found in the figure legends. Proportions of fly populations were tested with the Wilson-Brown method to determine 95% confidence intervals. For categorical data, such as mitochondria morphology scores, a Fisher's exact test was used to test for statistical significance between genotypes. Other quantitative data was assessed for normality using the Shapiro–Wilk test. For normally-distributed data, p-values were calculated using a one-way ANOVA test followed by Bonferroni's or Dunnett's multiple comparison tests. The Kruskal-Wallis test, with Dunn's multiple comparisons test, was used for non-parametric data sets. For multiple comparison tests, significance between groups was determined as adjusted p-value less than 0.05. For all experiments, no prior sample size estimation was performed. Sample sizes were determined from previous studies. For all experiments, the collection of subjects (flies) in each genotype was randomized, and no inclusion/exclusion was performed. When practical and necessary, blinding of genotypes to observer was performed. All data quantification was done in a blind or automated manner.

RStudio v.2022.07.2 running R v.4.2.2 was used for processing of RNA-seq data and generating the resulting plots. Details of the analysis pipeline are available in the previous

description of RNA Seq Analysis. Generally, significance was determined after Benjamini-Hochberg correction and at a level of adjusted p.value< 0.05. For microscopy experiments, raw Airyscan confocal images were acquired and processed in Zen Black (Zeiss). Images were analyzed in FIJI/ImageJ2 v.2.3.0 [78] and quantification was finished in Microsoft Excel and graphed with Graphpad Prism. Images and figures were arranged in either Microsoft Powerpoint or Inkspace (https://inkscape.org).

## Resource availability

All unique/stable reagents and animal stocks generated in this study are available from the lead contact and will be made available on request.

## Supporting information

**S1 Fig. Verification of *parkin* and *sting* mutant alleles.** Related to Fig 1.
(PDF)

**S2 Fig. Analysis and validation of an independent *sting*$^{\Delta RG5}$;*park*$^{25}$ stock.** Related to Fig 1.
(PDF)

**S3 Fig. Measurements of phosphorylated Ubiquitin and p62 from mutant thorax samples.** Related to Fig 4.
(PDF)

**S4 Fig. Analysis of Sting-regulated innate immunity in *park* mutants.** Related to Fig 5.
(PDF)

**S5 Fig. RNA-Seq experimental details and additional sample comparisons.** Related to Fig 5.
(PDF)

**S1 Table. All *D. melanogaster* genotypes, listed by figure.**
(DOCX)

**S1 Video. Geotaxis Assay Example—Used to quantify climbing activity in flies.** Related to Fig 1. Genotypes from left to right (note: vials were randomly assigned and blinded): D: *sting*$^{\Delta RG5}$; *park*$^1$, C: *sting*$^{\Delta RG5}$, B: *park*$^1$, A: *sting*$^{\Delta RG5}$; *park*$^{25}$, E: *w*$^{1118}$.
(MOV)

**S1 Data File. Data for Figure Graphs.xlsx—Includes all plotted data and statistics.**
(XLSX)

**S2 Data File. Appended Results file for RNA Sequencing.xlsx—Related to Fig 5.**
(XLSX)

## Acknowledgments

The authors thank Hong Xu (NHLBI, NIH), Rachel Cox (Uniformed Services University of the Health Sciences), and members of the Youle lab for feedback on the project and manuscript. Fly stocks were graciously provided by Alexander Whitworth (University of Cambridge), Leo Pallanck (University of Washington), Alicia Pickrell (Virginia Tech), Ed Giniger (NINDS, NIH), Jean-Luc Imler (Université de Strasbourg) and Alan Goodman (Washington State University). Dr. Helmut Krämer (UT Southwestern Medical Center) provided the polyclonal p62 antibody and valuable technical advice. Stocks obtained from the Bloomington

Drosophila Stock Center (NIH P40OD018537) were used in this study. STING plasmids were obtained from the Drosophila Genomics Resource Center (NIH 2P40OD010949). The microscopy experiments were supported by the NINDS Intramural core Light Imaging Facility (LIF).

## Author Contributions

**Conceptualization:** Andrew T. Moehlman, Richard J. Youle.

**Data curation:** Andrew T. Moehlman, Gil Kanfer.

**Formal analysis:** Andrew T. Moehlman, Gil Kanfer.

**Funding acquisition:** Richard J. Youle.

**Investigation:** Andrew T. Moehlman.

**Methodology:** Andrew T. Moehlman.

**Project administration:** Richard J. Youle.

**Resources:** Richard J. Youle.

**Software:** Gil Kanfer.

**Supervision:** Richard J. Youle.

**Validation:** Andrew T. Moehlman.

**Visualization:** Andrew T. Moehlman.

**Writing – original draft:** Andrew T. Moehlman.

**Writing – review & editing:** Andrew T. Moehlman, Gil Kanfer, Richard J. Youle.

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
