## [Decision Letter · Decision Letter 0]

24 Feb 2023

Dear Dr Youle,

Thank you very much for submitting your Research Article entitled 'Loss of *STING* in *parkin* mutant flies suppresses muscle defects and mitochondria damage' to PLOS Genetics.

The manuscript was fully evaluated at the editorial level and by independent peer reviewers. The reviewers appreciated the attention to an important problem, but raised some substantial concerns about the current manuscript. Based on the reviews, we will not be able to accept this version of the manuscript, but we would be willing to review a much-revised version. We cannot, of course, promise publication at that time.

If you decide to revise the manuscript for further consideration at PLOS Genetics, please aim to resubmit within the next 60 days, unless it will take extra time to address the concerns of the reviewers, in which case we would appreciate an expected resubmission date by email to plosgenetics@plos.org.

We are sorry that we cannot be more positive about your manuscript at this stage. Please do not hesitate to contact us if you have any concerns or questions.

Yours sincerely,

Bingwei Lu

Academic Editor

PLOS Genetics

Hua Tang

Section Editor

PLOS Genetics

Reviewer's Responses to Questions

**Comments to the Authors:**

Reviewer #1: I was excited when I first read this manuscript by Dr. Moehlman. In a way, it validates some unpublished phenomena we saw in pilot experiments (mentioned later). This manuscript attempts to address the interesting and critical question of whether cGAS/STING-mediated innate immune pathways affect the progression of PINK1-Parkin-associated Parkinson's disease. In mouse PD models, mitochondrial damage leads to the release of DAMPs that activate the cGAS/STING pathway. However, in the previous article published by Whitworth lab on the relationship between STING pathway and PINK1/Parkin (Lee et al., 2020, Sci Rep), they did not observe obvious genetic interaction. To be honest, we felt a little disappointed and confused. Because in our pilot experiments, we found in some pilot experiments that activating PP2A can maintain those dopaminergic neurons that would be degenerated in parkin mutants. And, in a recent paper by us and our collaborators (Ho et al., 2023, JCI), we found that PP2A inactivation can stimulate the activity of the cGAS/STING pathway and vice versa. It means, PP2A overexpression inhibits STING activity then rescues parkin. Given that both the Whitworth lab and the Youle lab have long-established reputations in mitophagy research, I think differences in genetic background are probably responsible for the observed differences in phenotypes.

Overall, I support the publication of this paper: on the one hand, this kind of academic debate is beneficial to the mitophagy society; on the other hand, this is a well-designed study, especially the genetics part (Fig. 1&2) . But I have a few questions and suggestions that I hope the author can consider as appropriate in their revision:

1) Does overexpression of STING in parkin mutants enhance the mutant phenotype and mitochondrial damage?

2) In addition to the signals already detected, have the authors thought about directly detecting DAMPs produced by mitochondria by using ddPCR? I think this should be a relatively direct evidence.

3) I think the determination of the GST pathway in Fig. 4 is a bit far-fetched. Of course, Fig.4A, 4B provide good support. But I feel that the 2-fold increase in GST activity shown in Fig. 4E does not necessarily provide a sufficient rationale for the previously observed rescues. In addition, if overexpressing GST or activating the GST pathway in parkin mutants, can defects be rescued as well as in STING KO? Or, if GST is knocked down in STING; park double mutant flies, will the previous rescue reverse? I think these genetic experiments can help to confirm whether GST is one of the most important downstream pathways activated.

4) I think the discussion of PINK1 in the article is not particularly sufficient, and I hope it can be strengthened.

I think this is a good manuscript and should be weighed positively.

Reviewer #2: This manuscript reports a genetic interaction between the Parkinson’s disease (PD) gene parkin and STING, which encodes a key component of an important evolutionarily conserved pathway of innate immunity. Parkin mediates selective autophagy of stressed mitochondria (mitophagy) and an important question in the field of PD relates to the consequences of the accumulation of damaged mitochondria in the absence of Parkin. One hypothesis that received experimental support is that stressed mitochondria release molecules that are sensed by the innate immune system and trigger inflammation. Indeed, this group previously reported that inhibition of STING in a mouse model deficient for parkin prevented inflammation, motor defect and neurodegeneration. In mammals, STING is activated by a cyclic dinucleotide second messenger produced by the enzyme cGAS, which acts as a receptor for cytosolic DNA, including released mitochondrial DNA. Here, the authors address the relevance of the interaction between parkin and STING in the drosophila model and report that, as in mice, loss of STING suppresses the muscle defects and mitochondria damage observed in parkin mutant flies. The results presented are convincing and novel. Of note, they contradict a previous study that concluded to a lack of interaction between parkin and STING. It is however not clear to me if this information is of sufficient novelty to justify publication in PLoS Genetics, especially given that some of the results are surprising (lack of interaction between pink1 and STING) and that the assets of the drosophila model are not really exploited to bring the field further in the understanding of the interaction between inflammation and PD. Addressing the comments listed below may make this manuscript a better candidate for publication in PLoS Genetics.

1) Lane 110: I note that a single allele of pink1 is used. Are other alleles available? My concern here is the issue of the genetic background masking the interaction with STING in one of the park[25] mutant lines. Has this pink1 line been isogenized in the same genetic background as the parkin mutants?

2) Fig. 3C: I find surprising that the amount of p-ubiquitin increases in sting; park double mutants. An easy control for the hypothesis put forward by the authors lane 163 would be to include a loading control with a mitochondrial protein.

3) It is unfortunate that double mutant fly lines for parkin and Relish could not be generated. The explanation proposed by the authors (lane 178), which I doubt after looking at the evidence presented in the two articles cited, could easily be tested by raising the flies in axenic conditions or on antibiotics. A genetic interaction between Relish and parkin, even if opposite to the one between sting and parkin, would be interesting. To confirm an involvement of STING signaling in the parkin phenotype, the authors should target another component of the pathway, such as IKKb.

4) Besides induction of a transcriptional response, STING can activate autophagy. Could a STING-dependent, possibly non canonical, autophagy explain the phenotype observed by the authors?

5) An obvious question raised by the data presented is whether the effect of STING is signal-dependent or constitutive? In other words, do cGAS-like receptors also contribute to the phenotype of parkin mutant flies? Is there an increase of cytosolic DNA or RNA in parkin mutant flies?

6) Minor comment: the order of the supplementary figures does not follow their appearance in the text (e.g., lane 163).

**Have all data underlying the figures and results presented in the manuscript been provided?**

Reviewer #1: Yes

Reviewer #2: Yes

PLOS authors have the option to publish the peer review history of their article (what does this mean?). If published, this will include your full peer review and any attached files.

Reviewer #1: **Yes: **Zhihao Wu

Reviewer #2: No

---

## [Decision Letter · Decision Letter 1]

13 Jun 2023

Dear Dr Youle,

We are pleased to inform you that your manuscript entitled "Loss of *STING* in *parkin* mutant flies suppresses muscle defects and mitochondria damage" has been editorially accepted for publication in PLOS Genetics. Congratulations!

Yours sincerely,

Bingwei Lu

Academic Editor

PLOS Genetics

Hua Tang

Section Editor

PLOS Genetics

Comments from the reviewers (if applicable):

Reviewer's Responses to Questions

**Comments to the Authors:**

Reviewer #1: My questions have been addressed. I support its publication.

Reviewer #2: I thank the authors for their effort in addressing the comments I raised. With the new information provided, this manuscript is a strong candidate for publication in PLoS Genetics.

**Have all data underlying the figures and results presented in the manuscript been provided?**

Reviewer #1: Yes

Reviewer #2: Yes

PLOS authors have the option to publish the peer review history of their article (what does this mean?). If published, this will include your full peer review and any attached files.

Reviewer #1: **Yes: **Zhihao Wu

Reviewer #2: No

**Data Deposition**

http://datadryad.org/submit?journalID=pgenetics&manu=PGENETICS-D-23-00084R1

**Press Queries**

---

## [Editor Report · Acceptance letter]

6 Jul 2023

PGENETICS-D-23-00084R1 

Loss of *STING* in *parkin* mutant flies suppresses muscle defects and mitochondria damage 

Dear Dr Youle, 

We are pleased to inform you that your manuscript entitled "Loss of *STING* in *parkin* mutant flies suppresses muscle defects and mitochondria damage" has been formally accepted for publication in PLOS Genetics! Your manuscript is now with our production department and you will be notified of the publication date in due course.

With kind regards,

Zsuzsanna Gémesi

PLOS Genetics

On behalf of:
